# Global DNA Methylation and Hydroxymethylation Levels in PBMCs Are Altered in RRMS Patients Treated with IFN-β and GA—A Preliminary Study

**DOI:** 10.3390/ijms24109074

**Published:** 2023-05-22

**Authors:** María Paulina Reyes-Mata, Mario Alberto Mireles-Ramírez, Christian Griñán-Ferré, Mercè Pallàs, Lenin Pavón, José de Jesús Guerrero-García, Daniel Ortuño-Sahagún

**Affiliations:** 1Laboratorio de Neuroinmunobiología Molecular, Instituto de Investigación en Ciencias Biomédicas (IICB), Centro Universitario de Ciencias de la Salud (CUCS), Universidad de Guadalajara, Guadalajara 44340, Mexico; 2Unidad Médica de Alta Especialidad (UMAE), Hospital de Especialidades (HE), Centro Médico Nacional de Occidente (CMNO), IMSS, Guadalajara 44340, Mexico; 3Pharmacology Section, Department of Pharmacology, Toxicology and Therapeutic Chemistry, Faculty of Pharmacy and Food Sciences, Institute of Neuroscience, Universitat de Barcelona, 08028 Barcelona, Spain; 4CiberNed, Network Center for Neurodegenerative Diseases, National Spanish Health Institute Carlos III, 28220 Madrid, Spain; 5Laboratorio de Psicoinmunología, Instituto Nacional de Psiquiatría Ramón de la Fuente Muñiz, Mexico City 14370, Mexico; 6Banco de Sangre Central, Unidad Médica de Alta Especialidad (UMAE), Hospital de Especialidades (HE), Centro Médico Nacional de Occidente (CMNO), IMSS, Guadalajara 44340, Mexico; 7Departamento de Farmacobiología, Centro Universitario de Ciencias Exactas e Ingenierías (CUCEI), Universidad de Guadalajara, Guadalajara 44340, Mexico

**Keywords:** multiple sclerosis (MS), epigenetic, DNA methylation, DNA hydromethylation, histone acetylation, Interferon beta (IFN-β) treatment, Glatiramer Acetate (GA) treatment

## Abstract

Multiple sclerosis (MS) is a chronic disease affecting the central nervous system (CNS) due to an autoimmune attack on axonal myelin sheaths. Epigenetics is an open research topic on MS, which has been investigated in search of biomarkers and treatment targets for this heterogeneous disease. In this study, we quantified global levels of epigenetic marks using an ELISA-like approach in Peripheral Blood Mononuclear Cells (PBMCs) from 52 patients with MS, treated with Interferon beta (IFN-β) and Glatiramer Acetate (GA) or untreated, and 30 healthy controls. We performed media comparisons and correlation analyses of these epigenetic markers with clinical variables in subgroups of patients and controls. We observed that DNA methylation (5-mC) decreased in treated patients compared with untreated and healthy controls. Moreover, 5-mC and hydroxymethylation (5-hmC) correlated with clinical variables. In contrast, histone H3 and H4 acetylation did not correlate with the disease variables considered. Globally quantified epigenetic DNA marks 5-mC and 5-hmC correlate with disease and were altered with treatment. However, to date, no biomarker has been identified that can predict the potential response to therapy before treatment initiation.

## 1. Introduction

Multiple sclerosis (MS) is a chronic disease affecting the central nervous system (CNS) that results from an autoimmune attack on axonal myelin sheaths [1,2]. MS has a heterogeneous character that is clearly manifested in its clinical forms [3]. Relapsing–remitting MS (RRMS) is the most common form (80–85% of cases), in which clinical activity occurs in the form of relapses followed by periods of remission with complete or partial recovery [4].

RRMS has an obvious gender dimorphism in terms of disease prevalence, clinical features, and disease progression [5]. Women are more commonly affected than men, with a ratio of 2 to 3:1 [6]. Currently, there is no cure for MS, and the use of Disease-Modifying Therapies (DMT) slows neurologic damage by reducing inflammation. However, treatment efficacy and radiologic and histopathologic changes in MS are heterogeneous, and patient follow-up is difficult [7].

First-line treatment includes Interferon beta (IFN-β) and Glatiramer Acetate (GA), which are gradually being replaced with newer treatments such as fingolimod and rituximab [8]. Because the efficacy of first-line therapy is low and a high percentage of patients do not respond to therapy [9], a different therapeutic regimen is then selected for patients; however, it is associated with a higher likelihood of side effects [8]. The search for biomarkers to predict treatment response and patient follow-up is one of the most important areas for the investigation of MS. Currently, some molecular biomarkers can be used as indicators for a poor response to DMT when patients are already on treatments [7]. Neutralizing antibodies to IFN-β, in addition to Myxovirus Resistance Gene A (*MxA*), have been recommended by the Neutralizing Antibodies on Interferon beta in Multiple Sclerosis consortium along with clinical and imaging studies to make individualized decisions about IFN-β treatment [10]. However, to date, this is not routinely considered in clinical practice [11]. Moreover, no biomarker predicts potential response to therapy before treatment initiation [12].

Epigenetic mechanisms regulate genes without altering the DNA sequence [13,14]. These mechanisms occur in neurodegeneration, inflammation, and immunological processes [15]. They also contribute to the heterogeneity of MS such as risk, evolution [16,17], and sex differences [18,19]. Thus, incomplete penetrance, epigenetic changes, and various environmental factors are responsible for susceptibility to the disease. Moreover, although twins have identical genotypes, they may observe different outcomes in their disease susceptibility. In fact, only one-third of monozygotic twins are concordant for MS, highlighting the role of epigenetics [20]. Scientific efforts have focused on the role of various epigenetic mechanisms involved in the pathophysiology of MS, but the information on peripheral blood is still insufficient.

First, there is evidence for differential methylation of several genes involved in the pathogenesis of MS [21,22,23,24,25,26,27]. The expression of DNA methyltransferase 1 (DNMT1) and Ten-Eleven Translocation enzyme 2 (TET2), which is involved in the conversion of 5-methylcitosine (5-mC) to 5-hydroxymethylcitosine (5-hmC), is decreased in PBMCs, and TET2 correlates with the time of disease evolution [28]. In this line, global 5-hmC levels are decreased [28], in contrast to global DNA methylation (5-mC), which is increased in blood cells from nontreated RRMS patients [29] and varies with patient age and sex [30]. Interestingly, there is evidence that DMT can alter 5-mC levels [30,31,32].

Second, there is a study on the degree of global histone acetylation in RRMS patients. In this study, the authors conclude that there are no differences between H3 and H4 acetylation in PBMC from RRMS patients and control subjects [33]. However, histone acetylation may modulate the immune system [34,35,36]. In this sense, to our knowledge, there is insufficient information on the global levels of 5-mC, 5-hmC, and histone acetylation in patients with MS with or without treatment and on whether these levels are related to clinical variables to understand if they can be used as biomarkers for diagnosis, treatment, or disease progression. Thus, global changes in epigenetic markers may influence disease progression and be associated with the clinic of MS patients [17].

This is an interesting phenomenon that can be associated with experimental approaches in which treatment with methylation inhibitors ameliorates the course of Experimental autoimmune encephalomyelitis (EAE) in mice and is a promising therapeutic strategy [37,38]. Therefore, global changes in epigenetic markers could alter the disease course and be associated with the clinic of MS patients.

In this study, we aim to investigate the possibility of an association between global levels of 5-mC, 5-hmC, and histone H3 and H4 acetylation in PBMCs with clinical variables from RRMS patients and healthy controls. H3 and H4 acetylation levels were poorly associated with disease in both patients and controls, but rather, were associated with the natural aging process, at least when considered globally. In contrast, 5-mC and 5-hmC correlated with clinical variables, and different 5-mC levels were found between treated and untreated patients. Further studies are needed to quantify global 5-mC and 5-hmC levels, especially with respect to treatment. These and other studies suggest that 5-mC may be related to treatment. However, a distinction must be made between methylation and hydroxymethylation to avoid bias in the results.

## 2. Results

### 2.1. Demographic and Clinical Data for the Patients

Healthy control subjects were matched with patients for sex and age, and no significant differences were found between control subjects and RRMS patients. There were also no significant differences between males and females in age, disease duration, Expanded Disability Status Scale (EDSS), Multiple Sclerosis Severity Score (MSSS), and Body Mass Index (BMI) among patients (Table 1) [39].

In agreement with previous findings from our laboratory on RRMS [39,40,41], it is clear that because of the heterogeneity of the disease, it is necessary to analyze epigenetic markers using subgroups of patients by treatment, BMI, sex, and duration of DMT use to better understand the pathophysiology of the disease MS.

Different subgroups of RRMS patients and healthy controls were classified according to clinical and demographic variables. Appendix A shows the significant correlations between the global values of the epigenetic markers and the clinical variables, including Spearman’s rank correlation coefficient, the *p*-value, and the sample size (n). The total number of subgroups that had significant correlations and the *p*-values for each variable interaction are shown in Appendix A. Some interactions were not significant in any subgroup, so they are shown as blanks. Finally, Appendix A shows the mean comparisons of the four epigenetic markers in the subgroups by sex. Representative plots showing the most important correlations and mean comparisons are provided below in Appendix A.

The results are presented from the less important to more important findings. In summary, histone acetylation was poorly correlated with clinical variables, in contrast to DNA methylation and hydroxymethylation.

### 2.2. The Quantitative Variables Correlated with Each Other in a Logical Manner

The correlation analysis revealed that the global levels of epigenetic markers in some subgroups correlated with each other (e.g., 5-mC vs. 5-hmC) (Appendix A shows the subgroups studied according to the different variables). Global histone H3 and H4 acetylation levels correlated positively with each other in several subgroups (31 subgroups). At the same time, 5-mC and 5-hmC also correlated positively with each other in several subgroups (8 subgroups). In two subgroups, 5-mC correlated negatively with histone H4 acetylation (Figure 1, Appendix A).

Several other clinical and demographic variables correlated in multiple subgroups due to their association with time: age, disease evolution, and timing of DMT use. In addition, the MSSS is calculated based on the EDSS and the timing of disease development, so a strong correlation between them is expected. These interactions can be found in Appendix A in the gray area.

### 2.3. Histone Acetylation Correlates with Age but Hardly with Clinical Variables in MS

Histone H3 but not H4 acetylation levels, as determined using optical density (O.D.), were statistically different between RRMS patients and healthy controls. Visually, however, this difference is not clear (Figure 2, Table 2). No differences were observed when patients were stratified by DMT (Table 2).

Global histone H3 and H4 acetylation levels were correlated with age in a total of 11 subgroups (Appendix A, dark blue area; Appendix A, panel a). We found a few correlations between histone acetylation and clinical variables: Histone H3 acetylation correlated with the time of disease evolution in two subgroups, and histone H4 acetylation correlated with EDSS and MSSS in a total of four subgroups (Appendix A, light blue area; Appendix A, panels e and f). Representative plots for the corresponding correlations are shown in Figure 3. Histone acetylation H3 and H4 correlated with BMI in only two subgroups, so their possible association with BMI is not striking (Appendix A, panel b).

### 2.4. The Correlation between DNA Hydroxymethylation (5-hmC) and Body Mass Index (BMI) in Healthy Controls Is No Longer Present in RRMS Patients

Global 5-hmC levels were directly associated with BMI in healthy controls, whereas hydroxymethylation was higher in healthy controls with higher weight, and 5-hmC correlated with BMI (Figure 4a,c, Appendix A, panel b). However, these correlations were lost in RRMS patients (Figure 4b,d), and only in GA-treated patients was there a negative correlation between these two variables (Appendix A, panel b).

### 2.5. Global DNA Methylation (5-mC) Is Decreased in RRMS Patients Compared with Healthy Controls

Global levels of epigenetic DNA markers in PBMCs (5-mC and 5-hmC) were compared between RRMS patients and healthy controls. Global 5-mC levels were significantly higher in healthy controls compared with RRMS patients (Figure 5a, Table 3), but no differences in DNA hydroxymethylation were observed (Figure 5b, Table 3). Furthermore, 5-mC also correlated with age in several subgroups (Appendix A, dark green area; Appendix A, panel a). However, it was also strongly associated with clinical variables (see below for more details). This may suggest that, in contrast to histone acetylation, 5-mC changes as a result of the pathophysiological process rather than the natural aging of patients.

### 2.6. Global DNA Methylation (5-mC) and DNA Hydroxymethylation (5-hmC) Vary Differently in RRMS Patients and Are Correlated with Clinical Variables

Global 5-mC levels decrease in treated patients compared with healthy controls, independently of DMT, and this decrease persists as a trend regardless of sex (Figure 6a, Appendix A). Strikingly, 5-hmC levels showed differences between patients and healthy controls only in males, in whom we observed a significant decrease in %5-hmC, whereas female patients resemble healthy controls (Figure 6b, Appendix A).

Global 5-mC and 5-hmC were correlated with all clinical variables in RRMS patients in different subgroups (Appendix A, light green area). Both 5-mC and 5-hmC correlated with disease evolution in patient groups, especially in subgroups with shorter disease evolution (Appendix A, area c and d). In the case of 5-hmC, it correlated with the timing of DMT use in overweight patients (Appendix A, panel d).

Global 5-mC levels correlated negatively with EDSS only in male patients (Figure 7a; Appendix A, panel e). Conversely, global 5-mC levels correlated positively with EDSS and MSSS only in female patients (Figure 7b,c; Appendix A, panels e and f). Other correlations were found for subgroups in which EDSS and MSSS were correlated with 5-mC only in males and with 5-hmC in females (Appendix A). In summary, 5-mC and 5-hmC behaved differently from the disability and severity scales when analyzed by sex.

### 2.7. Global Level of DNA Methylation in Peripheral Blood Mononuclear Cells (PBMCs) Is Decreased in Interferon Beta (IFN-β)- and Glatiramer Acetate (GA)-Treated Patients but Increased in Untreated Patients

We found a significant decrease in 5-mC in treated patients compared with untreated patients, even lower than in healthy controls (Figure 8a). Interestingly, the 5-mC reduction was observed in both IFN-β- and GA-treated patients compared with untreated patients (Figure 8b, Table 4). DMT was thus able to reduce 5-mC even to a lower level than in healthy controls (Figure 8a). We further investigated whether 5-mC varies as a function of the time of DMT intake. When we performed a correlation analysis, no correlation was seen in the whole group of patients. Remarkably, when we classified patients as less and more than 8 years by duration of the evolution, we observe a correlation between 5-mC and the duration of DMT use in treated patients with fewer than 8 years of evolution (Figure 8c; Appendix A, panel d). Furthermore, when considering patients with less duration of DMT use at different cut-off values, the percentage of 5-mC correlated with the duration of DMT use, with fewer than 5 years being the most significant point (Figure 8d; Appendix A, panel d). The difference between treated and untreated patients was observed only at 5-mC but not at 5-hmC (Table 4).

### 2.8. Participant Type (Patient vs. Control) and Age Are Variables Related to Low or High DNA Methylation Levels in Peripheral Blood Mononuclear Cells (PBMCs)

Logistic regression was performed to analyze the relationship between DNA methylation and participant type (patient vs. control) with age, BMI, and sex as confounding variables. The 75th percentile (21.7%) was chosen as the cutoff point for determining a high or low percentage of DNA methylation, i.e., samples with less than 21.7% DNA methylation were classified as low, and those with more than 21.7% were classified as high.

In the model, RRMS patients and younger participants had low levels of DNA methylation. BMI and gender had no statistical significance. Therefore, age and type of participants (patients vs. controls) were significant in explaining DNA methylation levels (Table 5). No other epigenetic characteristic showed a statistically significant result in the logistic regression.

## 3. Discussion

Epigenetic research in MS has increased in recent years to explain MS risk and heterogeneity [42] in the absence of complete genetic causality [43]. Research has focused mainly on posttranslational modification of 5-mC and histone and, more recently, 5-hmC [44]. In the present study, global levels of epigenetic markers were quantified in PBMCs from 57 RRMS patients and 31 healthy controls.

The global levels of epigenetic markers correlated with each other according to their biological function (Figure 1). Epigenetic modifications to DNA (methylation and hydroxymethylation) correlated positively with each other in eight subgroups. This is consistent with previous reports that methylation and hydroxymethylation can change in the same sense [45,46]; moreover, both 5-mC and 5-hmC are increased in PBMCs under the same circumstances [47,48]. Histone H3 and H4 were strongly and positively correlated in 31 subgroups. This is also consistent with previous reports that acetylation of histone H3 and H4 in blood cells moves in the same direction [49,50,51], probably because of the common goal of chromatin relaxation to allow gene expression. Finally, 5-mC correlated negatively with histone H4 acetylation in two subsets. In the promoter regions of genes, 5-mC is associated with gene repression, whereas histone acetylation is associated with gene transcription [52]. Therefore, the negative correlation found in these subgroups corresponds to their opposite function. However, most subgroups did not show these correlations, implying that multiple cell types, different gene expressions, and probably physiopathology could alter these interactions and, therefore, they cannot be seen globally.

Regarding histone acetylation, there was no difference between patients and healthy controls in global histone H4 acetylation (Figure 2b), and even when statistical analysis indicated a difference in histone H3 acetylation between patients and healthy controls, this difference was not visually clear (Figure 2a). Histone acetylation was negatively correlated with the timing of DMT application, EDSS, and MSSS in some subgroups of MS patients (Figure 3). In addition, both histone H3 and H4 were negatively correlated with age in seven subgroups of healthy controls and patients (Appendix A panel a, Figure 3). It has been reported that alteration in histone acetylation is involved in the aging process and that loss of histones is a common feature of aging in various living organisms including humans [53]. Therefore, the observed correlations with the time of disease evolution and the time of DMT use may be due to aging per se rather than the MS process. It has been previously reported that global histone acetylation in PBMCs from MS patients is not different from that in control groups [33]. Here, we confirmed that global acetylation is not strongly affected by either the MS process or treatment and may simply be related to aging in both MS patients and healthy controls. In contrast, a significant increase in histone H3 acetylation has been previously reported in the normal-appearing white matter (NAWM) of MS patients [54]. It is interesting to note how cell-specific epigenetic marks are, as they are involved in very different cellular processes: remyelination versus inflammation [55]. In summary, global histone acetylation in PBMCs does not differ between MS patients and control subjects and is not strongly correlated with clinical variables. Therefore, it might not be a good peripheral biomarker for MS. However, it is important to mention the possibility that some genomic sites might be differentially acetylated in MS compared with control subjects, which could influence disease or DMT action, even if these changes are not globally observable; this is a topic of interest for further studies.

DNA hydroxymethylation was positively correlated with BMI in subgroups of healthy controls (Figure 4a,c). Previously, 5-hmC was detected in adipose tissue of healthy controls, diabetics, and hypertensives [56] and in the placenta of obese women [57], but it seems that tissue-specific differences in this epigenetic marker cannot be fully compared because none of these studies yielded like ours. Samples used in these studies were placenta and adipose tissue instead of PBMC, and in all of them, 5-hmC was negatively correlated with obesity. In contrast, Nicoletti et al. (2016) quantified global 5-hmC in blood DNA from control subjects and obese women. In that study, global 5-hmC levels correlated positively with BMI and waist circumference. Interestingly, global 5-hmC levels decrease in obese women after bariatric surgery [58]. Therefore, blood-derived 5-hmC levels may be associated with obesity in healthy individuals, with higher global levels of 5-hmC as BMI increases for molecular mechanisms yet to be described. Contrary to 5-hmC in our study, 5-mC did not correlate with BMI. This also corresponds with Nicoletti et al., where Long Interspersed Element–1 (LINE-1) methylation was not correlated with BMI or waist circumference [58].

On the other hand, obesity has been described to play an important role in MS as in other autoimmune diseases by influencing disease progression, increasing severity, and affecting response to treatment. It has even been suggested that childhood obesity is a susceptibility factor that increases the risk of MS [59]. However, no correlation was found between hydroxymethylation and BMI in subgroups of RRMS patients (Figure 4b,d). Therefore, BMI might affect 5-hmC levels only in healthy subjects and not in patients, suggesting that there is a mechanism likely related to DMT intake that has not yet been described or to pathogenesis affecting DNA hydroxymethylation levels in PBMCs. In this regard, Calabrese found that total 5-hmC levels are reduced by the downregulation of TET2 enzyme expression in PBMCs from MS untreated patients [28], suggesting that pathogenesis per se affects blood 5-hmC levels. In mice, deletion of *Tet2* leads to alterations in 5-hmC and hypermethylation of *Foxp3*, impairing the differentiation and function of Treg cells, which are important for immune tolerance [2]. This hypothesis could be supported by the fact that global 5-hmC levels correlate with clinical variables in several subgroups of patients (Appendix A, panels c, d, e, and f). Other reports have pointed out that hydroxymethylation is affected in other tissues apart from blood. It is reduced in the spinal cord of EAE mice after induction of the model [60], and there are differentially hydroxymethylated sites in the DNA of postmortem isolated white matter neurons from MS patients compared to healthy controls [61]. In conclusion, hydroxymethylation is an epigenetic mark that remains to be explored in MS and may be related to disease progression and treatment. In addition, 5-hmC may be a future biomarker for pathogenesis or treatment that is not affected by patient BMI despite the known and recently explored the association between obesity and this epigenetic mark. Given the increasing importance of 5-hmC, a distinction from 5-mC should be considered when exploring the epigenetics of this disease [44].

BMI has been associated with the risk of MS [59,62,63], although the results regarding its association with brain volume, inflammation, and disability from MS are controversial [64,65,66]. In addition, an association between BMI and 5-mC was established by Castro et al. (2019) in MS, with DNA isolated from whole blood showing higher 5-mC in untreated MS patients [21]. In contrast, no correlation between BMI and 5-mC was found in any subgroup of treated MS patients in our study. Therefore, DMT must also affect the influence of BMI on 5-mC. Higher DNA methylation levels were observed in untreated patients compared with treated patients, although the small size of the group did not allow correlation analysis. Thus, DMT alters 5-mC, even in overweight and obese patients. Treatment heterogeneity in the samples of included patients may be an important variable for the questionable association between BMI and the progression of MS in this and other studies. Future studies need to decipher the molecular mechanism for the reduction in global methylation, its association with BMI, and its potential use as a biomarker.

We also found a sexual dimorphism in the 5-mC and 5-hmC profiles as a function of clinical variables. First, global 5-hmC values were higher in male controls than in male patients, whereas they were the same in female patients and healthy controls (Figure 6b). In Calabrese et al. (2104), it was described that global 5-hmC levels were reduced in PBMCs from MS untreated patients compared with healthy controls, most of whom were female [28]. In the present study, most patients were also female; however, most of them were treated patients. It is therefore likely that DMT affects 5-hmC in a sex-dependent manner, reversing the loss of 5-hmC in female patients and maintaining the differences between male patients and healthy controls [28]. In contrast, the mean comparison between global 5-hmC in patients and healthy controls showed a decrease independent of sex (Figure 6a), as did the comparison between the total patient and control groups (Figure 5a). In summary, 5-mC appears to be decreased in patients of both sexes compared with healthy controls, whereas 5-hmC is different in men but not in women, suggesting a possible biomarker for sexual dimorphism.

Second, we found an interesting phenomenon of sexual dimorphism (Figure 7): EDSS and MSSS correlated negatively with 5-mC in male patient subgroups, whereas these clinical variables correlated positively with 5-mC in female patient subgroups. Other female and male subgroups showed the same phenomena (Appendix A, panels e and f). Our results confirm these gender differences and may be indicative of epigenetics as a mechanism for the different outcomes of MS in women and men. Gender is a very important variable to consider when studying MS. The hypothesis for this difference according to sex is due to sex chromosomes, hormonal status, or a combination of both, which in turn leads to differences in epigenetic modulation of gene expression and its influence on immune response and neurobiology [18,67]. With our results, it becomes clearer that differential modulation of gene expression by epigenetic mechanisms is an important feature to understand and explain the sexual dimorphism that can also be observed in different biomolecules in the serum of RRMS patients [39,68].

In several studies, 5-mC has been associated with EDSS scores. For example, in Neven et al. (2016), a low EDSS score is associated with lower LINE-1 methylation in whole blood DNA. However, in the same group of patients, a higher EDSS score was associated with lower 5-mC in Alu sequences [69]. Therefore, different repetitive elements may be differentially affected by the disease. Interestingly, in Diniz et al. (2021), the EDSS score was positively associated with the global 5-mC level in monocytes from RRMS patients [32]. Consequently, the global 5-mC level may still shed light on exciting phenomena. In our study, the EDSS level did not correlate with the 5-mC level in the whole group. However, these variables were negatively correlated in the male subgroups of patients (Figure 7a), although correlations were not observed in any female group. In contrast, correlations with EDSS at 5-mC were observed in females (Figure 7b).

In the above studies, most of the patients were female, leading to results that at first glance do not agree with ours. However, Neven et al. used bisulfite sequencing of repetitive elements, which are known to detect both methylation and hydroxymethylation poorly [44]. Interestingly, in our study, DNA hydroxymethylation correlated positively with EDSS in the female subgroups (Figure 7b), suggesting that hydroxymethylation is the epigenetic mark that might be more affected by the pathophysiological process and/or treatment in females. This highlights the importance of the technical approach to identify the difference between methylation and hydroxymethylation. In addition, Diniz et al. used an ELISA-based method and calculated the relative percentage of global 5-mC concentration. However, the patient sample was less numerous and more heterogeneous in terms of treatment compared with ours [32]. Therefore, future studies need to be performed to analyze the relationship between EDSS and all these variables and the relationship between global 5-mC concentration and treatment. None of the aforementioned papers included the MSSS score in their analysis. However, in our study, the MSSS also correlated with 5-mC and 5-hmC in the same sense as the EDSS (Appendix A, panels e and f; Figure 7c). In conclusion, EDSS and MSSS correlate with global 5-mC and 5-hmC in a sex-dependent manner, and further studies are best suited to unravel the molecular mechanism and involvement of epigenetics in this sex dimorphism and the implications for different pathological outcomes.

The most important observation in our study is that global 5-mC was reduced in treated RRMS patients compared with both healthy controls and untreated RRMS patients (Figure 8a,b). The analysis of global methylation levels showed that they decreased in treated patients in contrast to untreated and clinically healthy subjects. However, no differences were observed between the two treatments, suggesting that the decrease in methylation levels is not a specific molecular mechanism of the two treatments but may be due to the change in immunological pro-inflammatory to anti-inflammatory profile induced with DMT per se. Interestingly, global 5-mC correlated negatively with the duration of DMT use in patients with disease evolution of fewer than 8 years and strongly with fewer than 5 years of DMT use (Figure 8c,d). Very recently, a longitudinal study showed an increase in 5-mC in untreated patients when comparing the first visit and follow-up. Moreover, patients on treatment (including IFN-β and GA) maintained hypomethylated sites in CD8+ lymphocytes compared with baseline [31]. Similarly, monocytes from IFN-β-treated patients were reported to have lower 5-mC levels than healthy controls, and a tendency toward the same phenomena was observed in fingolimod-treated patients [32]. In addition, it has been suggested that LINE-1 methylation levels may serve as a biomarker of treatment response during the first year of IFN-β treatment [30]. Another recent epigenome-wide association study highlighted that most of the differentially methylated genes tend to be hypomethylated in treated RRMS patients compared with controls, especially in enhancer regions. DMTs included in this study were dimethyl fumarate, GA, and IFN-β [27]. Therefore, global 5-mC is a potential biomarker for the treatment response that needs further exploration and motivates further research in this regard.

Although we cannot determine the exact type of mononuclear cells analyzed because it is a mixture, we know that the average percentage of mononuclear cell populations in peripheral blood is as follows: CD4+ T cells ~50%, CD8+ T cells ~20%, CD19+ B cells ~10%, CD16+ NK cells ~10%, CD16+ NK T cells ~10%, and CD16+ CD3+ cells ~10% [70,71]. In addition, it has been reported that RRMS patients have a higher proportion of CD4+ and CD8+ T cells, while the proportion of B cells remains the same compared to healthy individuals [72]. Therefore, our methylation data may mainly reflect CD4+ T helper and CD8+ cytotoxic lymphocytes.

In summary, hypomethylated states are associated with better patient outcomes. Inhibition of methylation has been investigated as a potential treatment for EAE. In studies conducted using this model, DNMT inhibitors were found to decrease the infiltration of lymphocytes into the CNS, leading to an increase in immunoregulatory cytokines and a decrease in those with inflammatory properties. In addition, an increase in Treg lymphocyte populations has been observed [37,38]. Thus, it appears that at less 5-mC by DNMTs, inflammation decreases. Experimental work and reports in humans, including the present report, agree that global hypomethylation of DNA in mononuclear cells is associated with favorable scenarios at MS/EAE. In Calabrese et al. (2014), the *TET2* gene was found to be hypermethylated and consequently less expressed in MS patients. Additionally, *TET2* expression correlated negatively with the disease duration [28]. The decreased expression of *TET2* in PBMCs might be related to lower demethylation because less 5-mC is converted to 5-hmC, resulting in a different pattern of global methylation and hydroxymethylation, which in turn might promote the inflammatory state in MS. Further studies need to be proposed to determine the cause of the differences in methylation/hydroxymethylation in immune cells from MS patients and how DMT may affect this status.

Interestingly, the correlation between the global 5-mC value and the time of DMT application occurred only in patients with a shorter disease course and a shorter time of DMT application. The cut-off values we found here are also consistent with our previous studies [39,40,41]. In addition, it has been described that the profile of cytokine expression shows greater fluctuations with a longer disease course [73] and that cytokine expression may also be affected by the transition from RRMS to Secondary Progressive MS (SPMS), with decreasing correlations between cytokines [74]. Therefore, changes in the physiopathology of MS, attributable to prolonged disease progression, must also affect 5-mC and the likely response to treatment, and vice versa.

Metagenome-wide association studies (MWAS) have shown that immune cells from MS patients have a different methylome than control subjects, with some genes repressed and others activated [21,22,23,24,25,26,75]. However, some reports, including this one, have highlighted that it is possible to observe interesting phenomena when the methylome is analyzed in a global manner. For example, PBMCs from untreated RRMS patients have higher global 5-mC compared with healthy controls [28,29]. For example, elevated 5-mC in LINE-1 is increased in RRMS patients compared with healthy controls [76], has been observed with higher EDSS [69], and has been associated with increased risk of relapse in total leukocytes (Area Under the Curve, AUC = 0.858) [30]. LINE-1 are repeated sequences throughout the genome used as a reference for the global status of 5-mC [69]. These results support further studies linking RRMS clinic to global DNA methylation and hydroxymethylation.

A logistic regression showed that DNA methylation could be explained by age and the type of participant (whether it is an RRMS patient or a control). It is interesting to note that patients and younger participants were associated with lower DNA methylation. As discussed above, higher DNA methylation is associated with MS, and the use of DMT decreases DNA methylation in immune cells [30,32,76]. Here, most of the patients were treated patients. Therefore, decreased DNA methylation can be associated with treatment use. It would be fascinating to increase the sample of untreated patients to observe differences with clinical variables inside the patient population and also to measure the clinical activity.

## 4. Materials and Methods

### 4.1. Patients and Healthy Controls

In total, 57 patients (33 women and 24 men) diagnosed with RRMS were recruited from the Neurological Service of the Western National Medical Center of the Mexican Institute of Social Security (IMSS) in Jalisco, Mexico. The inclusion and exclusion criteria for RRMS patients were established in our previous report [39]: (1) diagnosis of RRMS according to the revised McDonald diagnostic criteria (2017) [77]; (2) age between 20 and 60 years; and (3) RRMS who had either been treated with IFN-β (*n* = 26) or GA (*n* = 26) for at least 3 months or had not received any treatment for at least 2 years (*n* = 5). Clinical disability was assessed using Kurtzke’s Expanded Disability Status Scale (EDSS) [78], and disease severity was assessed using the Multiple Sclerosis severity score (MSSS) [79]. The clinical form of RRMS was determined according to the revised criteria of Lublin and Reingold [80]. The control group consisted of 31 healthy subjects (21 women and 10 men) matched for age and sex and selected from the central blood bank of the Western National Medical Center’s Specialties Hospital at IMSS. Participants’ demographic and clinical data were obtained from hospital records or using a questionnaire. Samples were collected at the same time of day to avoid bias due to the circadian cycle. This study was conducted in accordance with the ethical guidelines of the 2013 Declaration of Helsinki and approved by the Ethics and Investigation Committees of the IMSS in Mexico (R-2019-785-066). All participants gave written informed consent to participate in the study.

### 4.2. Blood Sample Collection and Peripheral Blood Mononuclear Cells (PBMC) Isolation

Peripheral blood samples were collected in Vacutainer^®^ tubes (BD, New York City, NY, USA) containing ethylenediaminetetraacetic acid (EDTA) to obtain PBMCs. PBMCs were collected by density gradient centrifugation using lymphoprep™ (STEMCELL Technologies, Vancouver, BC, Canada) according to the manufacturer’s instructions. Further, after DNA and histone extraction, global levels of epigenetic marks were quantified. The term “global level” refers to the total amount of the epigenetic marks in the sample [81,82], which in this study was obtained using an ELISA-based method. The methodology is described in the following sections.

### 4.3. Quantification of Global DNA Methylation and Hydroxymethylation

Freshly isolated PBMCs were reconstituted in a 10% (*v*/*v*) solution of DNA/RNA shield™ (Zymo Research™, Irvin, CA, USA) for further extraction of DNA and stored at −80 °C. DNA was extracted using the FitAmp™ Blood and Cultured Cell DNA Extraction Kit (Epigentek, Farmingdale, NY, USA). DNA extracts were used for quantification of methylation and hydromethylation using the Methylflash™ Methylated DNA Quantification Kit (Epigentek, Farmingdale, NY, USA) and the Methylflash™ Hydroxymethylated DNA Quantification Kit (Epigentek, Farmingdale, NY, USA). Briefly, 100 ng of sample DNA was immobilized on a 96-well plate with high DNA affinity. Methylated DNA and hydroxymethylated DNA were then detected using capture and detection antibodies against 5-mC and 5-hmC. The plate was read at 450 nm using the FlexA-200 microplate reader (Allsheng, Hangzhou, China). The absolute percentage of methylated or hydroxymethylated DNA was calculated from the optical density (O.D.) using the slope of a standard curve (50% methylated DNA or 25% hydroxymethylated DNA).

### 4.4. Quantification of Global Histone H3 and H4 Acetylation

Histone H3 and H4 acetylation were quantified using the EpiQuik™ Global Histone H3 Acetylation Assay Kit (Epigentek, Farmingdale, NY, USA) and the EpiQuik™ Global Histone H4 Acetylation Assay Kit (Epigentek, Farmingdale, NY, USA) according to the manufacturer’s instructions. First, histones were isolated from freshly isolated PBMCs using the reagents of the respective kit as well as trichloroacetic acid (TCA) and acetone. Histones were then quantified using the Pierce™ Coomassie Plus Bradford Assay (Thermo Scientific™, Waltham, MA, USA). Histone extracts were stored at −80 °C until assayed. Subsequently, 1 ug of protein was immobilized on a 96-well plate and acetylated H3/H4 was detected with capture and detection antibodies. The plate was read at 450 nm using the FlexA-200 microplate reader (Allsheng, Hangzhou, China), and OD was displayed directly.

### 4.5. Statistical Analysis

A database was created using Microsoft Office Excel, and statistical analysis was performed using GraphPad Prism ver. 7.0. The mean (M) and standard deviation (SD) of the global values of epigenetic markers were calculated. The statistical outliers were removed after the analysis using the Grubbs test with a significance level of 5% [83]. According to the normality test, all these variables were not normally distributed, so nonparametric tests were performed: U Mann–Whitney test for comparison of means between two groups and Kruskal–Wallis, followed by Duns post hoc test for comparison of means between three or more groups. The Spearman r (rho) test was used to analyze correlations between epigenetic markers and clinical variables. Finally, a logistic regression was performed to assess the variables possibly involved in high or low levels of epigenetic markers, in which the confounder variables included were age, BMI, and gender. When *p* < 0.05, statistical significance was assumed.

## 5. Conclusions

The study of epigenetics in MS is an increasingly important research topic in the search for biomarkers and therapeutic targets. We found that histone H3–H4 acetylation was virtually unaffected compared with healthy controls, at least at a global level in PBMCs. Contrastingly, 5-mC and 5-hmC correlated with patients’ clinical variables, and 5-mC was different depending on the treatment. This study confirms that 5-mC and 5-hmC are epigenetic markers whose changes can be observed globally in MS patients taking DMT, although a cause-effect relationship remains to be discovered.

Globally quantified epigenetic changes in DNA may be an important biomarker in the future. However, longitudinal studies and patient follow-up are needed to assess the association and further clinical utility. Finally, a causal relationship needs to be demonstrated with experimental approaches. Nonetheless, global quantification of 5-mC and 5-hmC is an accessible tool that is attractive because of the possibility of observing global changes in the entire pool of PBMCs in this clinically associated disease. Finally, these results open a new avenue not only to understand the pathology and progression of the disease but also to develop new therapies based on epigenetics.

## Figures and Tables

**Figure 1 ijms-24-09074-f001:**
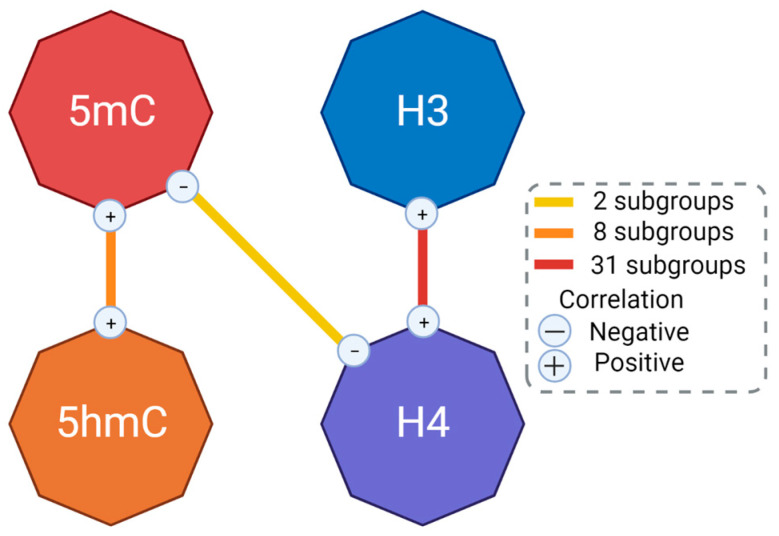
The number of correlations between global levels of epigenetic markers. The box shows the number of subgroups in which a particular interaction was found. H3, histone H3 acetylation; H4, histone H4 acetylation; 5-mC, DNA methylation; 5-hmC, DNA hydroxymethylation. Created with BioRender.com.

**Figure 2 ijms-24-09074-f002:**
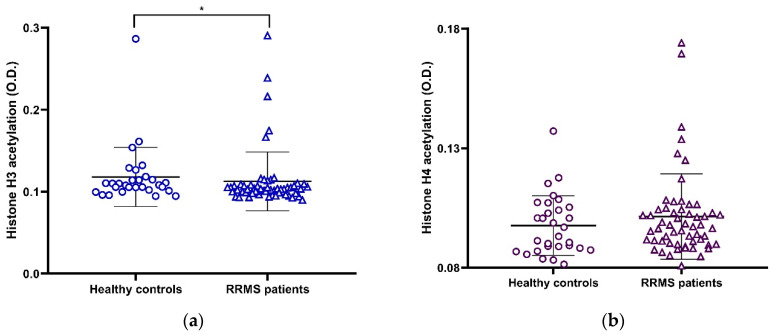
Global expression of (**a**) histone H3 acetylation, and (**b**) histone H4 acetylation between RRMS patients and healthy controls. (* *p* < 0.05, U Mann–Whitney); M ± SD. RRMS, Relapsing–Remitting Multiple Sclerosis; O.D., optical density. The symbols in the graph are just the same and represent individual determinations.

**Figure 3 ijms-24-09074-f003:**
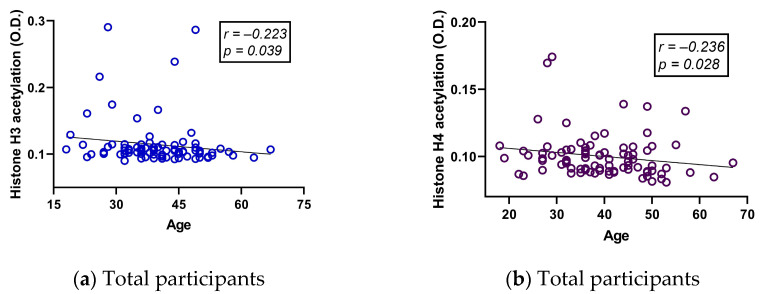
Representative correlations between variables and global values of histone H3 and H4 acetylation: (**a**) H3 acetylation vs. age in total participants, (**b**) H4 acetylation vs. age in total participants, (**c**) H4 acetylation vs. EDSS in normal weight patients, and (**d**) H4 acetylation vs. MSSS in normal weight patients. r, Spearman’s correlation coefficient; O.D., Optical Density; EDSS, Expanded Disability Status Scale; MSSS, Multiple Sclerosis Severity Score.

**Figure 4 ijms-24-09074-f004:**
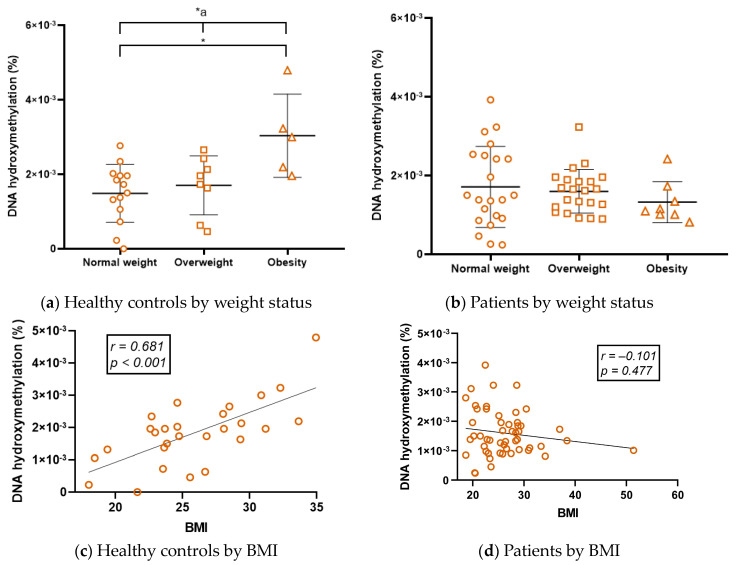
Differences according to weight status in global DNA hydroxymethylation within the (**a**) healthy control group and (**b**) RRMS patient group. Correlation between global DNA hydroxymethylation and BMI in the (**c**) healthy control group and the (**d**) RRMS patient group. M ± SD (*^a^ *p* ≤ 0.05, Kruskal–Wallis; * *p* < 0.05, Dunn’s test). BMI, Body Mass Index. r, Spearman’s correlation coefficient; RRMS, Relapsing–Remitting Multiple Sclerosis. The symbols in the graph are just the same and represent individual determinations.

**Figure 5 ijms-24-09074-f005:**
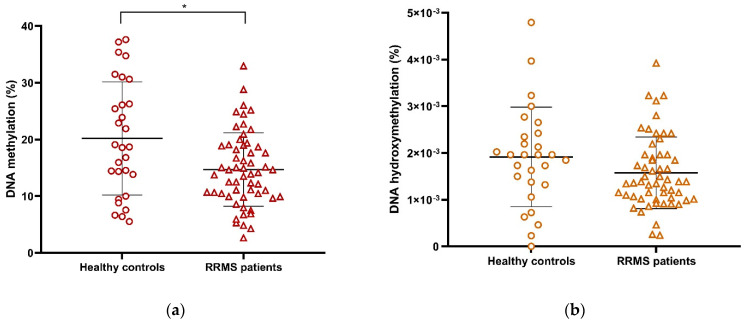
Global expression of (**a**) %5-mC and (**b**) %5-hmC between RRMS patients and healthy controls. (* *p* < 0.05, U Mann–Whitney); M ± SD. RRMS, Relapsing–Remitting Multiple Sclerosis; O.D., optical density; %5-mC, percentage of global DNA 5-methylcytosine; %5-hmC, percentage of global DNA 5-hydroxymethylcytosine. The symbols in the graph are just the same and represent individual determinations.

**Figure 6 ijms-24-09074-f006:**
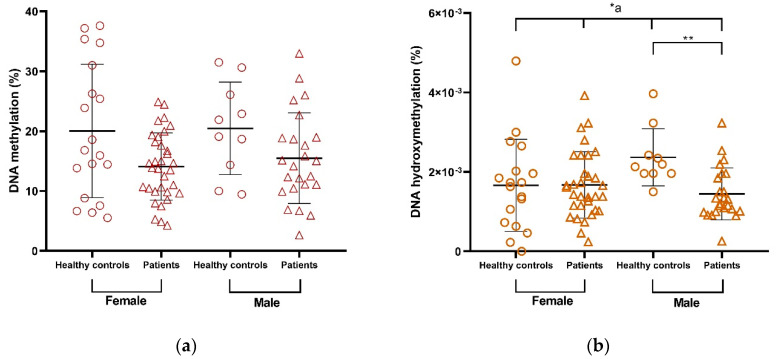
Differences in (**a**) %5-mC and (**b**) %5-hmC between RRMS patients and healthy controls stratified by sex. M ± SD (*^a^ *p* ≤ 0.05, Kruskal–Wallis test; ** *p* < 0.01, Dunn’s test). %5-mC, percentage of global DNA 5-methylcytosine; %5-hmC, percentage of global DNA 5-hydroxymethylcytosine. The symbols in the graph are just the same and represent individual determinations.

**Figure 7 ijms-24-09074-f007:**
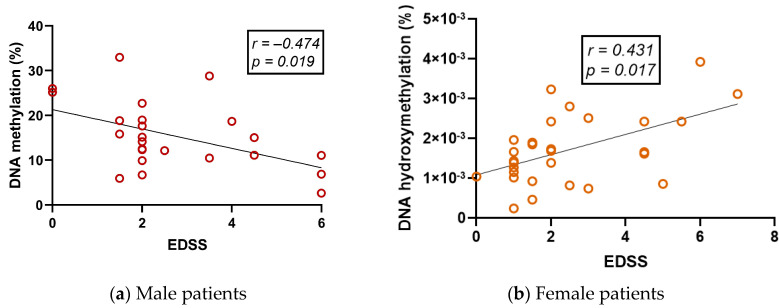
Correlations between EDSS and MSSS and global levels of DNA epigenetic markers: (**a**) %5-mC vs. EDSS in male patients, (**b**) %5-hmC vs. EDSS in female patients, and (**c**) 5-hmC vs. MSSS in female patients. EDSS, Expanded Disability Scale; MSSS, Multiple Sclerosis Severity Scale; r, Spearman’s correlation coefficient; %5-mC, percentage of global DNA 5-methylcytosine; %5-hmC, percentage of global DNA 5-hydroxymethylcytosine.

**Figure 8 ijms-24-09074-f008:**
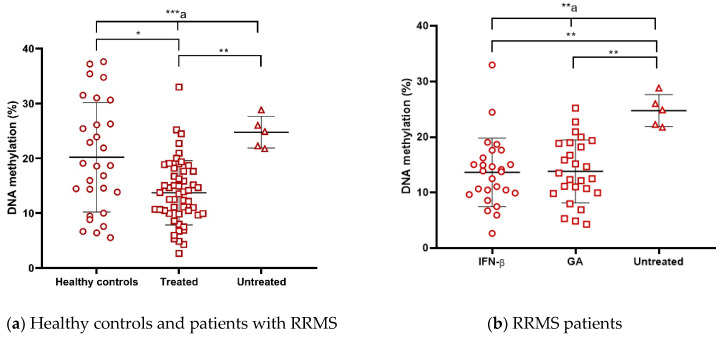
%5-mC of (**a**) healthy controls compared to treated and untreated patients and (**b**) RRMS patients stratified by DMT used. Correlation between %5-mC and the time of DMT use in (**c**) patients with fewer than 8 years of disease evolution and (**d**) fewer than 5 years of DMT use. M ± SD (***^a^ *p* < 0.001, **^a^ *p* ≤ 0.01, Kruskal–Wallis; ** *p* < 0.01, * *p* < 0.05, Dunn’s test). r, Spearman’s correlation coefficient; DMT, Disease-Modifying Therapy; RRMS, Relapsing–Remitting Multiple Sclerosis; %5-mC, percentage of global DNA 5-methylcytosine. The symbols in the graph are just the same and represent individual determinations.

**Table 1 ijms-24-09074-t001:** Demographic and clinical data for the patients with relapsing–remitting multiple sclerosis (RRMS).

N	Female	Male	Total	*p*
	33	24	57	-
(%)	60	40	100	-
Age (years)	40.9 ± 9.7	39.4 ± 9.3	40.3 ± 9.5	0.567 ^a^
Disease duration (years)	10.7 ± 6.3	10.0 ± 7.3	10.4 ± 6.6	0.389 ^b^
EDSS (*n* = 54)	2.4 ± 1.8	2.7 ± 1.7	2.5 ± 1.7	0.511 ^b^
MSSS (*n* = 54)	3.0 ± 2.9	3.7 ± 2.4	3.3 ± 2.7	0.282 ^b^
BMI (*n* = 53)	25.3 ± 4.6	27.2 ± 6.9	26.2 ± 5.7	0.107 ^b^

Data are expressed as Mean (M) ± Standard Deviation (SD). ^a^ Student’s *t* test and ^b^ U Mann–Whitney. EDSS, Expanded Disability Status Scale; MSSS, Multiple Sclerosis Severity Score; BMI, Body Mass Index.

**Table 2 ijms-24-09074-t002:** Comparison of global histone acetylation levels.

Variable	Subgroups	n	M ± SD	*p*
AcH3 (O.D.)	RRMS patients	57	0.113 ± 0.04	0.017 *
	Healthy controls	29	0.118 ± 0.04	
AcH3 (O.D.)	IFN-β patients	25	0.107 ± 0.02	0.618 ^a^
	GA patients	26	0.116 ± 0.04	
	Non-treated patients	4	0.102 ± 0.01	
AcH4 (O.D.)	RRMS patients	57	0.101 ± 0.02	0.377
	Healthy controls	30	0.098 ± 0.01	
AcH4 (O.D.)	IFN-β patients	25	0.01 ± 0.02	0.418 ^a^
	GA patients	26	0.103 ± 0.02	
	Non-treated patients	6	0.102 ± 0.01	

Data are expressed as Mean (M) ± Standard Deviation (SD). (* *p* < 0.05, U Mann–Whitney; ^a^ Kruskal–Wallis); O.D., optical density; AcH3, Global histones H3 acetylation; AcH4, Global histones H4 acetylation; RRMS, Relapsing–Remitting Multiple Sclerosis; IFN-β, Interferon beta; GA, Glatiramer Acetate.

**Table 3 ijms-24-09074-t003:** Comparison of DNA epigenetic marker levels between RRMS patients and healthy controls.

Variable	Subgroup	*n*	M ± SD	*p*
%5-mC	RRMS patients	57	14.69 ± 6.47	0.019 *
	Healthy controls	29	20.19 ± 9.97	
%5-hmC	RRMS patients	56	0.0016 ± 0.0008	0.075
	Healthy controls	28	0.0018 ± 0.0011	

Data are expressed as Mean (M) ± Standard Deviation (SD). (* *p* < 0.05, U Mann–Whitney). %5-mC, percentage of global DNA 5-methylcytosine; %5-hmC, percentage of global DNA 5-hydroxymethylcytosine; RRMS, Relapsing–Remitting Multiple Sclerosis.

**Table 4 ijms-24-09074-t004:** Comparison between epigenetic DNA marker levels in RRMS patients stratified by DMT.

Variable	Subgroups	*n*	M ± SD	*p*
%5-mC	IFN-β patients	26	13.63 ± 6.16 †	0.004 **
	GA patients	26	13.81 ± 5.67 ‡	
	Non-treated patients	5	24.75 ± 2.89 †‡	
%5-hmC	IFN-β patients	25	0.0015 ± 0.0008	0.418
	GA patients	26	0.0016 ± 0.0008	
	Non-treated patients	6	0.0016 ± 0.0006	

Data are expressed as M ± SD. The %5-mC had a statistical difference between subgroups (** *p* ≤ 0.01, Kruskal–Wallis). The difference was found between IFN-β-treated patients vs untreated patients (†) and between AG-treated patients vs. untreated patients (‡) (** *p* < 0.01, Dunn’s test). %5-mC, percentage of global DNA 5-methylcytosine; %5-hmC, percentage of global DNA 5-hydroxymethylcytosine; DMT, Disease-Modifying Therapy; RRMS, Relapsing–Remitting Multiple Sclerosis; IFN-β, Interferon beta; GA, Glatiramer Acetate.

**Table 5 ijms-24-09074-t005:** Association between variables and levels of DNA methylation.

Dependent Variable	Independent Variable	OR	IC 95%	*p*
%5-mC (high vs. low)	Control vs. RRMS Patient	5.189	1.608–16.743	0.006 *
	Age	0.899	0.838–0.964	0.003 *
	Gender	1.682	0.513–5.513	0.390
	BMI	1.030	0.923–1.149	0.597

Logistic regression. * *p* ≤ 0.01. OR, Odds ratio; CI, Confidence of interval; %5-mC, percentage of global DNA 5-methylcytosine; RRMS, Relapsing–Remitting Multiple Sclerosis; BMI, Body Mass Index.

## Data Availability

The data presented in this study are available on request from the corresponding author. The data are not publicly available due to privacy and ethical restrictions.

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
