# Peer review of "Global DNA Methylation and Hydroxymethylation Levels in PBMCs Are Altered in RRMS Patients Treated with IFN-β and GA—A Preliminary Study"

_ijms, 2023, doi:10.3390/ijms24109074_

Round 1

Reviewer 1 Report

Authors submit the manuscript entitle 'Global DNA methylation and hydroxymethylation levels in PBMC are altered in RRMS patients treated with IFN-β and GA',first the title is not suitable and unclear in keypoint. Please delete the word of global, your research is not global data. 2nd, you only use PBMC, you cannot delineate what immune cells are the essentail methylated. And in page 2 3rd line, the sentence is not easy to understand [Nevertheless, the efficacy of first-line therapy is low because a high percentage of patients do not respond to therapy (9), and the relapse rate is lower compared with other treatments (8).]

The fundemental problem is that your result cannot delineate the differnt of methylation who received either INF-b or GA. 

And DNA methylation and hydroxymethylation ha been reported a lot in obesity issue, (. Obes Sug 2016 Mar;26(3):603-11.) What''s impact of your result from these articles in obesity, like figure 4? Please discuss further.

Author Response

Response to Reviewer #1 Comments

Point 1: first the title is not suitable and unclear in keypoint. Please delete the word of global, your research is not global data.

Response 1: We think that we understand and share the reviewer’s point in the general sense. However, in this case, the term "global" in the title refers to the quantification of the total content of 5-methylcytosine (5-mC) and 5-hydroxymethylcytosine (5-hmC) in all DNA obtained from PBMCs and not to the standard of methylation or hydroxymethylation at a particular site or to global data from other analyzes. For this purpose, colorimetric quantification methods were used, employing optimized antibodies and enhancer solutions that prevent the occurrence of cross-reactions. These methods are explained in the relevant section of the manuscript and their advantages as a global methylation analysis are best explained by Li and Tollefsbol, 2021 [81]. Similarly, the term "global" has already been used by several authors in their studies to refer to the overall content of these epigenetic modifications, using different technical approaches [doi: 10.3390/toxics10040157], [doi: 10.1155/2015/845041], including ELISA-based methods [doi: 10.1371/journal.pone.0152849],[82]. In the Methods section, an explanation was added to clarify better that the term "global methylation" refers to the total amount of epigenetic marks found in the sample, as opposed to site-specific epigenetic modifications (new lines 528-532).

Point 2: 2nd, you only use PBMC, you cannot delineate what immune cells are the essentail methylated.

Response 2: We thank the reviewer for the opportunity to further explain this aspect. As mentioned in the previous comment, the purpose of the study was to measure total methylation and hydroxymethylation in PBMC from peripheral blood, which is referred to as "global". The study does not aim to comprehensively define the exact type of mononuclear cells analyzed because it is a mixture. However, although we cannot define the exact mononuclear cell type analyzed, as they are a mixture, we know that the average percentage of mononuclear cell populations in peripheral blood is as follows: CD4+ T cells ~50%, CD8+ T cells ~20%, CD19+ B cells ~10%, CD16+ NK cells ~10%, CD16+ NK T cells ~10%, CD16+ CD3+ cells ~10% [70], [71]. In addition, it has been reported that RRMS patients have a higher proportion of CD4+ and CD8+ T cells, while the proportion of B cells remains the same compared to healthy individuals [72]. Therefore, our methylation data may mainly reflect T-helper CD4+ and cytotoxic CD8+ lymphocytes. This information has been added to the manuscript in new lines 450-457.

Point 3: And in page 2 3rd line, the sentence is not easy to understand [Nevertheless, the efficacy of first-line therapy is low because a high percentage of patients do not respond to therapy (9), and the relapse rate is lower compared with other treatments (8).].

Response 3: We agree with the reviewer’s comment. Consequently the sentence was rewrited in new lines 51-53, as follows: "Because the efficacy of first-line therapy is low and a high percentage of patients do not respond to therapy (9), a different therapeutic regimen is then selected for patients. However, it is associated with a higher likelihood of side effects (8)."

Point 4: The fundemental problem is that your result cannot delineate the differnt of methylation who received either INF-b or GA.

Response 4: The reviewer's comment about the lack of differences in methylation between the two treatments is correct. However, we do not consider this to be a drawback of our study. On the contrary, our results demonstrate differences between treated and untreated patients and healthy subjects. The discussion has been improved by a statement highlighting this (new lines 430-435). In addition, there are reports in which several DMT reduce global DNA methylation in immune cells. These references are already mentioned in the Discussion.

Point 5: And DNA methylation and hydroxymethylation ha been reported a lot in obesity issue, (. Obes Sug 2016 Mar;26(3):603-11.) What''s impact of your result from these articles in obesity, like figure 4? Please discuss further.

Response 5: We thank the reviewer for pointing this out and for the opportunity to further explain this aspect. The reference suggested by the reviewer has already been cited in the discussion (Nicoletti, 2016). Our main idea is that there is a positive relationship between 5-hmc and obesity in healthy individuals in blood-derived DNA (described by Nicoletti and observed in Fig. 4 panels a and c in our study). This association is no longer present in RRMS patients (Fig. 4, panels b and d). It has been previously described that 5-hmC is impaired in PBMC and neurons from patients with multiple sclerosis, so we hypothesized that pathogenesis affects hydroxymethylation despite obesity. In agreement with the reviewer’s comment, the text in this section has been improved and a new reference has been added to better explain the possible molecular pathway for 5-hmC reduction in MS patients (new lines 325-327). This finding is important because 5-hmC has not yet been studied in depth in MS. If pathogenesis can affect it despite its relationship to obesity, it could be used as a biomarker in the future. This has also been added to the respective section (new lines 333-345, and 353-357).

Finally, we would like to thank the reviewers for their detailed revision of our manuscript and for their very constructive and much-appreciated comments.

Reviewer 2 Report

I have reviewed the paper by Reyes-Mata et al. which measures and compares global histone acetylation and methylation levels of the PBMCs derived from healthy donors and MS patients. The authors aim to find biomarkers that can be associated with MS treatment response and clinical variables. The paper reports significant associations between pairs of variables, including a negative association between DNA methylation and MS treatment, both IFN-beta or glatiramer acetate, and treatment duration.

However, my main concern relates to the statistical analysis and experimental design. The authors present a large number of tests that contrast pairs of variables in multiple subsets of samples. Most subgroups presented in Table S1 lack statistical power for statistical analysis, as the cells rarely present more than 5 individuals. As such, it is difficult to assess the significance of the reported associations.

Moreover, probably as a consequence of the limited sample size, the authors limit their analyses to bivariate non-parametric tests. The main drawback of this approach is that one cannot assess the contribution of other variables to explain a given association. For example, what is the significance of the association between methylation and treatment once corrected by factors such as BMI (which the authors show is highly associated with methylation), age, and sex in a multivariate analysis?

To address these concerns, the authors should present a table with the complete set of tests, bivariate and multivariate, indicating coefficients and P-values. This will enable a better understanding of the basis of all their results and conclusions.

In summary, while the paper reports significant associations between pairs of variables, the statistical analysis and experimental design could be improved. By presenting a more detailed analysis, the authors could strengthen the validity and impact of their findings.

Minor:

TableS2: Replace Spanish word “tiempo”.

TableS3: P > 0.001 be replaced by p < 0.001.

Author Response

Response to Reviewer #2 Comments

Point 1: However, my main concern relates to the statistical analysis and experimental design. The authors present a large number of tests that contrast pairs of variables in multiple subsets of samples. Most subgroups presented in Table S1 lack statistical power for statistical analysis, as the cells rarely present more than 5 individuals. As such, it is difficult to assess the significance of the reported associations.

Response 1: We thank the reviewer for his detailed scrutiny of the statistical analysis. First, we would like to apologize to the reviewer for the confusion caused by our supplemental Tables 1 and 2. The numbers reported in the main cells of these tables do not represent the "n" of each subgroup, but rather the number of significant correlations found in each subgroup. Because tables S1, S2, and S3 contained similar information about the same analysis performed, although the type of summary was different. However, based on the reviewer's comment, we reevaluated the information in each table and the common misconceptions when reading the tables. Consequently, we restructured the information in new supplemental tables (described in new lines 12-128, and 134-135).

Additionally, based on the reviewer's comment about the large number of tests contrasting pairs of variables in multiple subsets of samples and his suggestion, we conducted a multivariate analysis that led us to add a new Section 2.8 in addition to our original analysis. Here we added a multivariate analysis. We chose logistic regression because linear regression with a non-normally distributed variable is not feasible in the case of our epigenetic quantifications, and also because Pinto-Medel et al. made a similar approximation [30]. The sample size of untreated patients is too small, so we did not perform multivariate analysis in this case, hoping that shortly we can obtain a more significant number of untreated patients to observe the epigenetic marks concerning treatment and other relevant clinical variables. Nevertheless, we believe that bivariate analysis is sufficient to observe associations that are also consistent with previous reports. This encourages us to continue this line of research.

Point 2: Moreover, probably as a consequence of the limited sample size, the authors limit their analyses to bivariate non-parametric tests. The main drawback of this approach is that one cannot assess the contribution of other variables to explain a given association. For example, what is the significance of the association between methylation and treatment once corrected by factors such as BMI (which the authors show is highly associated with methylation), age, and sex in a multivariate analysis?

Response 2: We thank the reviewer for his suggestion and the opportunity to develop this aspect further. In line with the reviewer's suggestion, we have now included a multivariate analysis (mentioned in the statistical analysis in new lines 569-571) to better assess the relative contribution of each variable in explaining the low or high values of the epigenetic markers. This is now explained in the new section 2.8 of the results. In addition, the result of this analysis has been included in the new Table 5.

Point 3: To address these concerns, the authors should present a table with the complete set of tests, bivariate and multivariate, indicating coefficients and P-values. This will enable a better understanding of the basis of all their results and conclusions.

Response 3: We thank the reviewer for his suggestions on the statistical analysis and hope that the new tables with the "p" and "n" values for each comparison and the inclusion of the results of the multivariate analysis have met the reviewer's requirements. We are happy to further improve these analyzes as needed.

Finally, we would like to thank the reviewers for their detailed revision of our manuscript and for their very constructive and much-appreciated comments.

Round 2

Reviewer 2 Report

The authors addressed my main concerns.